# Core promoter factor TAF9B regulates neuronal gene expression

**Francisco J Herrera[1,2,3]\*, Teppei Yamaguchi[1,2,3], Henk Roelink[1], Robert Tjian[1,2,3]**

[1]Department of Molecular and Cell Biology, University of California, Berkeley, Berkeley, United States; [2]Howard Hughes Medical Institute, University of California, Berkeley, Berkeley, United States; [3]CIRM Center of Excellence, Li Ka Shing Center For Biomedical and Health Sciences, University of California, Berkeley, Berkeley, United States

**Abstract** Emerging evidence points to an unexpected diversification of core promoter recognition complexes that serve as important regulators of cell-type specific gene transcription. Here, we report that the orphan TBP-associated factor TAF9B is selectively up-regulated upon in vitro motor neuron differentiation, and is required for the transcriptional induction of specific neuronal genes, while dispensable for global gene expression in murine ES cells. TAF9B binds to both promoters and distal enhancers of neuronal genes, partially co-localizing at binding sites of OLIG2, a key activator of motor neuron differentiation. Surprisingly, in this neuronal context TAF9B becomes preferentially associated with PCAF rather than the canonical TFIID complex. Analysis of dissected spinal column from *Taf9b* KO mice confirmed that TAF9B also regulates neuronal gene transcription in vivo. Our findings suggest that alternative core promoter complexes may provide a key mechanism to lock in and maintain specific transcriptional programs in terminally differentiated cell types.

## Introduction

Sequence-specific transcription factors are essential for directing cellular differentiation and identity. These factors play key roles in establishing gene expression patterns that dictate cell-type specific functions and characteristics. Cellular reprogramming and de-differentiation leading to the formation of induced pluripotent stem cells offer dramatic examples of how targeted expression of particular sets of transcription factors can control cell fate (reviewed by *Graf, 2011*). Sequence-specific transcription factors work in concert with a cohort of co-activator complexes and core promoter recognition factors to execute transcriptional activation. These core promoter factors often called 'basal' or 'general' transcription factors have traditionally been considered invariant and universally required for the expression of all protein-coding genes. However, recent studies suggest that at least some components of this large and diverse group of factors necessary to form the transcriptional pre-initiation complex (PIC) can exhibit both promoter and enhancer targeting activities that are cell-type specific. This specificity depends on the variegated composition of the components in the PIC (reviewed in *D'Alessio et al., 2009*; *Goodrich and Tjian, 2010*; *Müller et al., 2010*).

The first evidence of tissue-specific functions of core PIC factors came from studies of TAF (TBP-associated factors) subunits in the core promoter recognition factor TFIID. A paralog of TAF4, called TAF4B, was found to control oocyte-specific activation of transcription in mouse ovaries (*Freiman et al., 2001*). In *Drosophila,* a group of five TAF paralogs (No hitter/TAF4; Cannonball/TAF5; Meiois I arrest/TAF6; Spermatocyte arrest/TAF8; and Ryan express/TAF12) all play specific roles in spermatogenesis (*Hiller et al., 2004*; *Chen et al., 2005*). Similarly, another orphan TAF, TAF7L, cooperates with TBP-related factor 2 (TRF2) to regulate spermatogenesis in mice (*Cheng et al., 2007*; *Zhou et al.,*

**\*For correspondence:** fjherrera@berkeley.edu

**Reviewing editor**: Robb Krumlauf, Stowers Institute for Medical Research, United States

**eLife digest** Almost all the cells in an organism contain the same genetic information, but they develop into many different types of cells that perform a variety of specialized functions in the body. Brain cells, for example, have a very different shape and function from red blood cells. A small group of proteins act inside cells to switch on the expression of genes it needs to carry out the specific functions of a given cell-type, and switch off the genes that are only needed in other cell types.

Some of these regulatory proteins called 'core promoter factors' bind to the DNA near the start of genes. These core factors are known to work in combination with various other proteins to switch genes on or off in specific cell types. However, the specific core promoter factors and partner proteins that guide a cell into becoming a neuron have not been well characterized.

Now, Herrera et al. have identified a core promoter factor called TAF9B that is produced at higher levels when mouse stem cells are coaxed into becoming the motor neurons that carry nerve impulses to muscles. The TAF9B protein works together with an enzyme (called PCAF) to help to switch on the genes that control the development of these cells. Without this regulatory protein, mouse stem cells grown in the lab fail to properly switch on the genes that are necessary to become motor neurons. These mutant stem cells also fail to efficiently switch off genes that stop stem cells from becoming more specialized. High levels of TAF9B were also found in the spinal cord of newborn mice and when Herrera et al. engineered mice that lack TAF9B, these mice did not properly regulate the expression of neuronal genes in their spines.

These new findings might, in the future, improve our ability to guide stem cells into forming neurons, or to reprogram other types of specialized cells into becoming motor neurons. This new information could also prove useful for researchers interested in better understanding neuronal development and might aid in the design of therapies to treat neuronal injuries or diseases, such as motor neuron disease.

2013a). Tissue-specific functions of TAF7L were also found in adipocytes where it acts in conjunction with PPARγ to control the transcription necessary for adipogenesis (*Zhou et al., 2013b*). In mouse embryonic stem (ES) cells, TAF3 pairs up with CTCF to drive the expression of endoderm specific genes while in myoblasts TAF3 works with TRF3 in the differentiation of myotubes (*Deato and Tjian, 2007*; *Liu et al., 2011*). Collectively these experiments suggest that combinations of different subunits of the multi-protein core promoter factors can be enlisted to participate in gene- and tissue-specific regulatory functions. Thus, mouse ES cells and other progenitor cells very likely have quite different requirements for such factors compared to terminally differentiated mature cell-types. Dissecting the various diversified mechanisms that control gene transcription in terminally differentiated cells should contribute to our still rudimentary understanding of the gene regulatory processes that modulate homeostasis in somatic cells and those that could lead to degeneration of adult tissue in disease states. A more detailed analysis of these critical molecular mechanisms may also help improve new strategies to achieve efficient cellular reprogramming and stem cell differentiation.

Despite emerging evidence for unexpected activities carried out by core promoter factors in various cellular differentiation pathways, little was known about their potential involvement in the formation of neurons during embryogenesis. In this study we explore whether TAFs or other core promoter recognition factors become engaged in neuronal specific functions to regulate the expression of neuronal genes. To address this question we used an in vitro differentiation protocol to induce murine ES cells to form spinal cord motor neurons (MN), which control muscle movement. Loss of motor neurons gives rise to devastating diseases, including amyotrophic lateral sclerosis (ALS) (reviewed by *Robberecht and Philips, 2013*). Consequently, motor neurons have been the focus of intense study and several key classical sequence-specific DNA-binding transcription factors regulating the expression of motor neuron-specific genes have been identified (reviewed by *di Sanguinetto et al., 2008*; *Kanning et al., 2010*). However, there was scant information regarding the role, if any, of core promoter factors in directing the network of gene transcription necessary to form neurons. In this report, we have combined genomics, biochemical assays, and gene knockout strategies to dissect the transcriptional mechanism used to generate motor neurons from murine ES cells in vitro as well as to uncover novel in vivo neuronal-specific changes in core promoter factor involvement and previously undetected co-activator functions.

## Results

### TAF9B is up-regulated upon neuronal differentiation

To examine whether the expression of various components of the core promoter recognition complex changes upon neuronal differentiation, we induced ES cells to form motor neurons using retinoic acid (RA) and the smoothened agonist SAG as described previously (*Wichterle et al., 2002*). We confirmed the generation of motor neurons in embryoid bodies (EBs) by immunostaining for motor neuron-specific markers LHX3 and ISL1/2 (*Figure 1A*) as well as by RNA-seq analysis (*Figure 1—figure supplement 1A*). To obtain enriched populations of motor neurons, we differentiated a murine ES cell line containing a motor neuron-specific promoter (*Mnx1*) fused to GFP as a means of isolating post-mitotic motor neurons present in these EBs (*Figure 1B*, *Figure 1—figure supplement 1B*; *Wichterle et al., 2002*). Using this highly enriched motor neuron cell population, we compared the mRNA expression levels of genes coding for canonical TAF subunits in TFIID as well as various TAF paralogs (*Figure 1C*). As seen in other differentiated cell types, several TAFs are down-regulated upon motor neuron differentiation (*Figure 1*; *Deato and Tjian, 2007*; *D'Alessio et al., 2011*; *Zhou et al., 2013b*; *Guermah et al., 2003*). Intriguingly, one TAF paralog (*Taf9b*) was significantly up-regulated in GFP

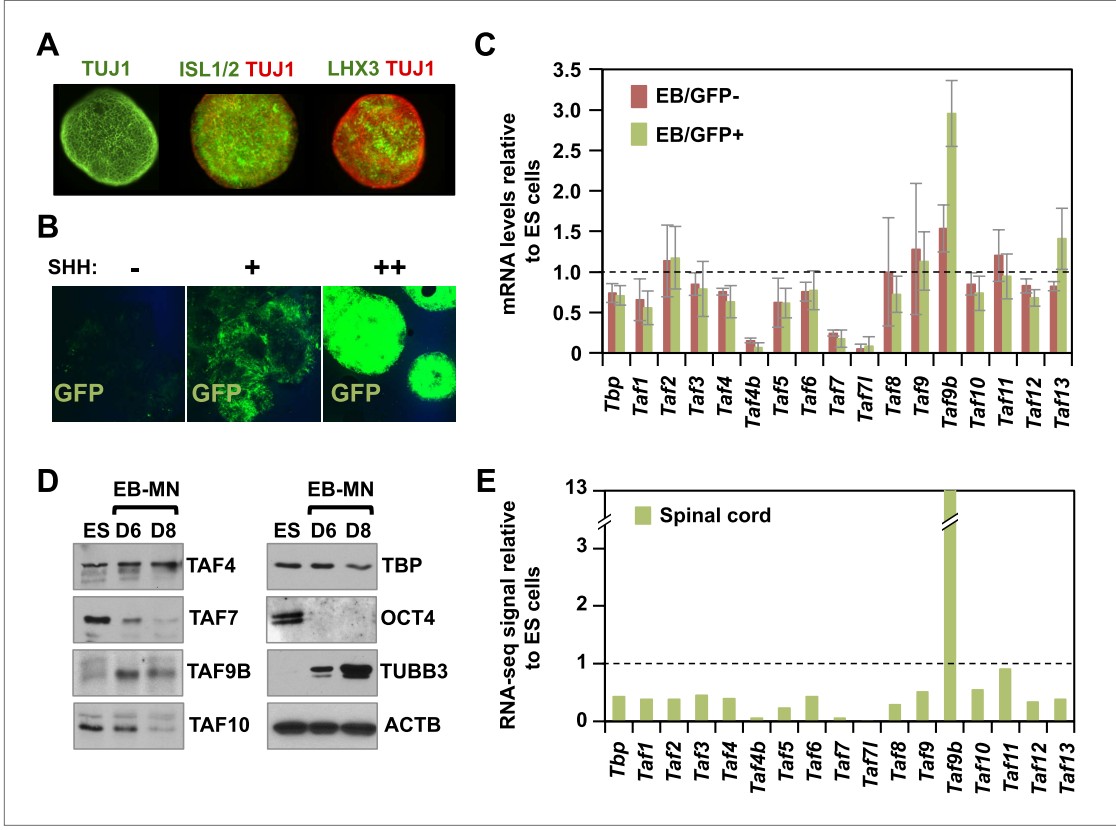

**Figure 1**. TAF9B is up-regulated upon neuronal differentiation. (**A**) Mouse embryonic stem (ES) cells were differentiated into motor neurons (MN) using embryoid body cultures (EB-MN) for 6 days and immunostained with MN markers ISL1/2 and LHX3, and neuronal marker TUJ1. (**B**) Mouse ES cells containing a MN-specific GFP marker were used to isolate GFP expressing (GFP+) MN cells and to monitor induction of MN under different sonic hedgehog (SHH) concentrations. (**C**) TAF expression levels were measured by qRT-PCR in GFP+ and GFP negative (GFP−) cells sorted from EB-MN differentiated for 6 days and compared to the levels observed in undifferentiated ES cells. Bars show mean ± SD of three biological replicates. (**D**) Western blot analysis of ES cells and EB-MN differentiated for 6 and 8 days detecting TAFs, the neuronal marker TUBB3, the ES cell marker OCT4, and ACTB. (**E**) RNA-seq analysis of TAF expression comparing mouse ES cells and mouse newborn spinal cord tissue. Bars show FPKM values of spinal cord tissue relative to FPKM values of ES cells.

The following figure supplements are available for figure 1:

**Figure supplement 1**. TAF9B is up-regulated upon neuronal differentiation.

expressing (GFP+) motor neurons but not in the GFP negative (GFP−) cells (*Figure 1C*). We confirmed that the up-regulation of TAF9B and down-regulation of several other TAFs occur at the level of both mRNA and protein (*Figure 1D*), and follows the kinetics of post-mitotic neuronal marker *Tubb3* but not the progenitor cell markers *Pax6* and *Olig2* (*Figure 1—figure supplement 1C*). We next dissected spinal cord tissue from newborn mice and performed RNA-seq to measure in vivo *Taf* expression levels and compare them to those observed for mouse ES cells in culture. As expected, most *Taf* subunits of TFIID in newborn spinal cord are expressed at lower levels than in mouse ES cells, while *Taf9b* is up-regulated more than 10-fold, consistent with the results obtained with the in vitro differentiated motor neurons (*Figure 1E*). Notably, changes in the expression levels of *Tafs* in newborn spinal cord are more pronounced than what we observed for the in vitro differentiated motor neurons. We also found that many components of the PIC and selected co-activators were down-regulated upon neuronal differentiation (*Figure 1—figure supplement 1D and 1E*). These results strongly suggest that induction of TAF9B upon neuronal differentiation is specific and distinct from regulated expression of most other subunits of the PIC.

## TAF9B is dispensable for global gene expression in mouse ES cells

To test the function of TAF9B in mouse ES cells and during motor neuron differentiation, we used *Taf9b* knock-out (KO) ES cells generated by the trans-NIH Knock-Out Mouse Project (KOMP). In these cells, all protein-coding exons of *Taf9b* have been replaced with the *LacZ-Neo* cassette (*Figure 2A*). As expected, the induction of *Taf9b* upon motor neuron differentiation was detected only in wild type (WT) cells but not in *Taf9b* KO cells (*Figure 2B,C*). We found that *Taf9b* KO cells have a normal mouse ES cell morphology and can be grown in the presence of LIF without displaying any obvious cell growth or cell cycle defects (*Figure 2D* and data not shown). To determine the role of TAF9B in regulating transcription in murine ES cells, we performed RNA-seq analysis to compare global gene expression

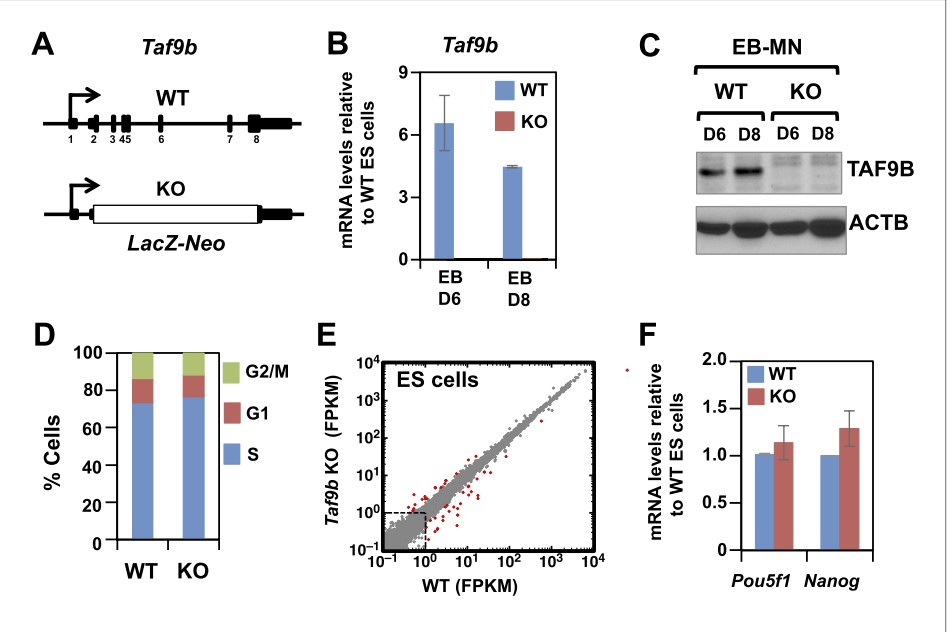

**Figure 2**. TAF9B is dispensable for global gene expression in murine ES cells. (**A**) In *Taf9b* KO murine ES cells a *LacZ-Neo* cassette replaces all protein coding exons of *Taf9b* gene on the X chromosome. (**B**) qRT-PCR of in vitro differentiated EB-MN for 6 and 8 days detecting the expression of *Taf9b* in WT and *Taf9b* KO cells. Bars show mean ± SEM of three biological replicates. (**C**) Western blot analysis of in vitro differentiated EB-MN for 6 and 8 days detecting the expression of TAF9B in WT and *Taf9b* KO cells. (**D**) Cell cycle profile of WT and *Taf9b* KO ES cells based on BrdU incorporation. Graph represents average values of two independent biological duplicates. (**E**) RNA-seq analysis of WT and *Taf9b* KO ES cells grown in undifferentiated conditions. Scatter plot represents FPKM values for genes expressed in WT and *Taf9b* KO cells. Red dots are genes whose expression is affected more than twofold with p-values <0.05. Gray dots represent genes not changing more than the selected cutoff. (**F**) qRT-PCR analysis of the ES cell markers *Pou5f1* and *Nanog* in WT and *Taf9b* KO ES cells grown in undifferentiated conditions. Bars show mean ± SEM of three biological replicates.

patterns in WT and *Taf9b* KO cells. The results show that TAF9B is largely dispensable for global gene expression with only a small number of genes deregulated under these conditions (*Figure 2E*). Interestingly, whereas the core TAF components of TFIID have recently been reported to be critical for murine ES cell pluripotency and in controlling genes such as *Pou5f1*, *Nanog* and *Klf4* (*Pijnappel et al., 2013*), the absence of TAF9B did not affect expression of pluripotency markers (*Figure 2E,F*). These results suggest that, unlike other TAFs present in the canonical TFIID complex, TAF9B is dispensable for global gene expression and pluripotency of murine ES cells.

## TAF9B is required for efficient differentiation of motor neurons

We next tested whether TAF9B is required for efficient in vitro motor neuron differentiation. ES cells were differentiated as described previously and motor neuron markers *Isl1*, *Mnx1*, and *Lhx3* were measured by qRT-PCR. We observed a significant reduction in all motor neuron markers in *Taf9b* KO cells upon differentiation (*Figure 3A*). Immunostaining for motor neuron marker ISL1/2 as well as western blot analysis for TUBB3 further confirmed that motor neuron differentiation was compromised

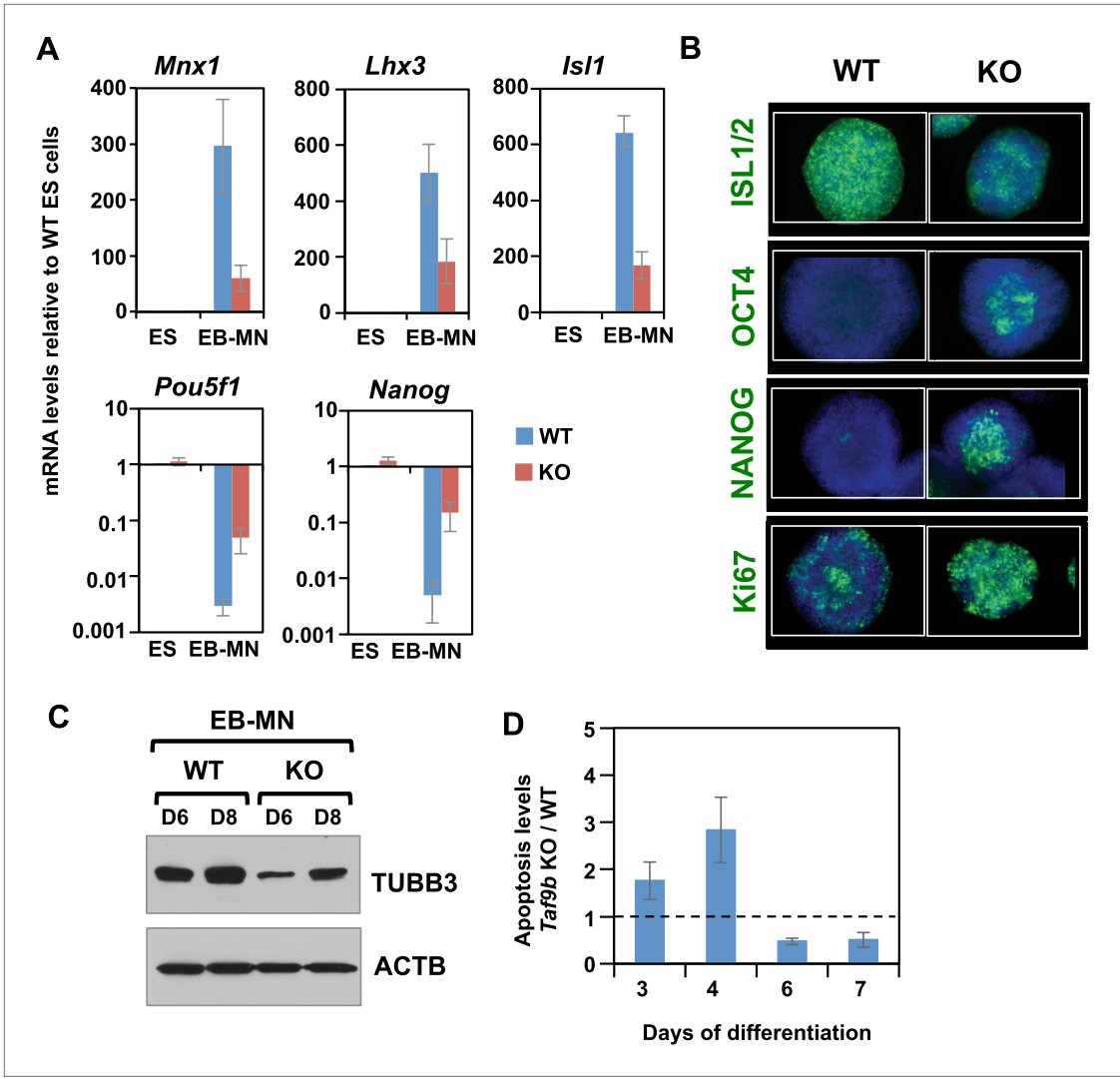

**Figure 3**. TAF9B is required for efficient differentiation of murine ES cells into motor neurons in vitro. (**A**) qRT-PCR analysis for the MN markers (*Mnx1*, *Lhx3*, *Isl1*) and ES cell markers (*Pou5f1* and *Nanog*) in WT and *Taf9b* KO cells differentiated for 6 days. Values are relative to undifferentiated ES cells. Bars show mean ± SEM of three biological replicates. (**B**) WT and *Taf9b* KO EB-MN, differentiated for 6 days, were immunostained using antibodies against MN marker ISL1/2, ES cell markers OCT4 and NANOG, and cellular proliferation marker Ki67. (**C**) Western blot analysis of WT and *Taf9b* KO EB-MN samples differentiated for 6 and 8 days detecting neuronal marker TUBB3 and ACTB. (**D**) Apoptosis levels were determined using an annexin V based assay in WT and *Taf9b* KO cells at different time points during MN differentiation. Graph shows mean ± SEM for the ratio of *Taf9b* KO to WT cells in three biological replicates.

in *Taf9b* KO cells (**Figure 3B,C**). We next measured the down-regulation of ES specific markers *Pou5f1* and *Nanog* upon differentiation and found that *Taf9b* KO cells failed to efficiently down-regulate these ES cell markers (**Figure 3A**). Immunostaining of OCT4 and NANOG confirmed that a fraction of cells in these neuralized EBs derived from *Taf9b* KO cells continue to express pluripotency markers, whereas few or no cells expressing OCT4 and NANOG were detected in WT cells (**Figure 3B**). Consistent with the maintenance of an undifferentiated state, a higher number of dividing cells persisted in *Taf9b* KO EBs as determined by the marker for cellular proliferation Ki67 (**Figure 3B**). We also found an increase in apoptosis in *Taf9b* KO cells compared to WT controls during early stages of differentiation (**Figure 3D**). These differences in apoptosis were particularly prominent at day 4 when progenitor cells are most enriched. This is consistent with previous observations implicating TAF9B in cell survival and control of apoptosis in human HeLa cells (***Chen and Manley, 2003***; ***Frontini et al., 2005***). Taken in aggregate, our findings suggest that TAF9B is required for the efficient in vitro differentiation of murine ES cells into motor neurons, and that the absence of TAF9B causes the persistent expression of pluripotency markers under differentiation conditions.

## Efficient induction of neuronal genes requires TAF9B

To obtain a more comprehensive view of the role of TAF9B during motor neuron differentiation, we performed RNA-seq analysis at regular intervals during in vitro motor neuron differentiation of WT and *Taf9b* KO ES cells. We found that as motor neuron differentiation progressed there was a concomitant increase in the number of genes deregulated by the loss of TAF9B (**Figure 4A**). Striking deregulation of genes in *Taf9b* KO cells was observed at the time point at which progenitor cells are most enriched in these cultures (day 4). Genes known to be important for neuronal progenitor identity such as *Olig2*, *Pax6*, *Sox1*, *Nkx6-1*, *Nkx6-2*, *Gli1*, and genes controlled by the Notch signaling pathway were among the most down-regulated by the loss of TAF9B (**Figure 4B**, **Figure 4—figure supplement 1A**). Gene ontology (GO) terms analysis of down-regulated genes showed categories such as neuron differentiation and anterior/posterior pattern formation among the most enriched terms (**Figure 4—figure supplement 1B**). Among up-regulated genes we found genes involved in apoptosis (***Supplementary file 1***), which is consistent with the increase in apoptosis levels we observed in *Taf9b* KO cells at this time point. By day 6, when post-mitotic motor neurons are normally present in these cultures, approximately 10% of all expressed genes were down-regulated and another 10% were up-regulated in *Taf9b* KO cells. This finding suggests that TAF9B is specifically required for the appropriate expression of a subset of genes, rather than for global gene expression in these differentiated EBs. The most significant GO term categories of genes down-regulated by the loss of TAF9B in this time point included genes involved in neuron differentiation, axogenesis, and neuron projection (**Figure 4C**). Moreover, the specific motor neuron markers *Lhx3*, *Lhx4*, *Isl1*, *Mnx1*, and pan-neuronal marker *Tubb3* were among the most down-regulated genes in *Taf9b* KO cells (**Figure 4B**). In contrast, genes up-regulated in *Taf9b* KO cells were enriched in categories such as vascular development, stem cell maintenance, and other non-neuronal developmental categories (**Figure 4C**). Contrary to the large number of genes deregulated in neuralized EB samples, we found that samples differentiated for 2 days have a relatively small number of genes affected in *Taf9b* KO cells. Interestingly, genes involved in differentiation were still found among these early down-regulated genes (***Supplementary file 1***). These results indicate that TAF9B controls the expression of specific genes required at least for the in vitro differentiation of murine ES cells into motor neurons.

## TAF9B binds promoters and distal regions of neuronal genes

To determine whether TAF9B acts directly to regulate neuronal genes we mapped its binding sites genome-wide by ChIP-seq. Because most TAFs are expected to bind to promoters as part of TFIID in the PIC, we compared ChIP-seq peaks of TAF9B with those of RNA POL2. Approximately a third of all detected TAF9B binding regions overlapped with RNA POL2 occupancy (TAF9B-POL2) and mapped generally near transcription start sites (TSS) of protein coding genes (**Figure 5A**, **Figure 5—figure supplement 1B,C**). However, TAF9B peaks located at TSS were not enriched at promoters of neuronal specific genes (**Figure 5B**). Intriguingly, approximately two thirds of all TAF9B binding regions showed little or no overlap with regions of high RNA POL2 occupancy (TAF9B-only) and were generally located distal to annotated TSS (**Figure 5A**, **Figure 5—figure supplement 1B, 1C**). Importantly, genes with TAF9B bound to regions distal to their TSS were significantly enriched in neuronal genes categories, including GO terms such as midbrain-hindbrain boundary development, rostrocaudal neural tube

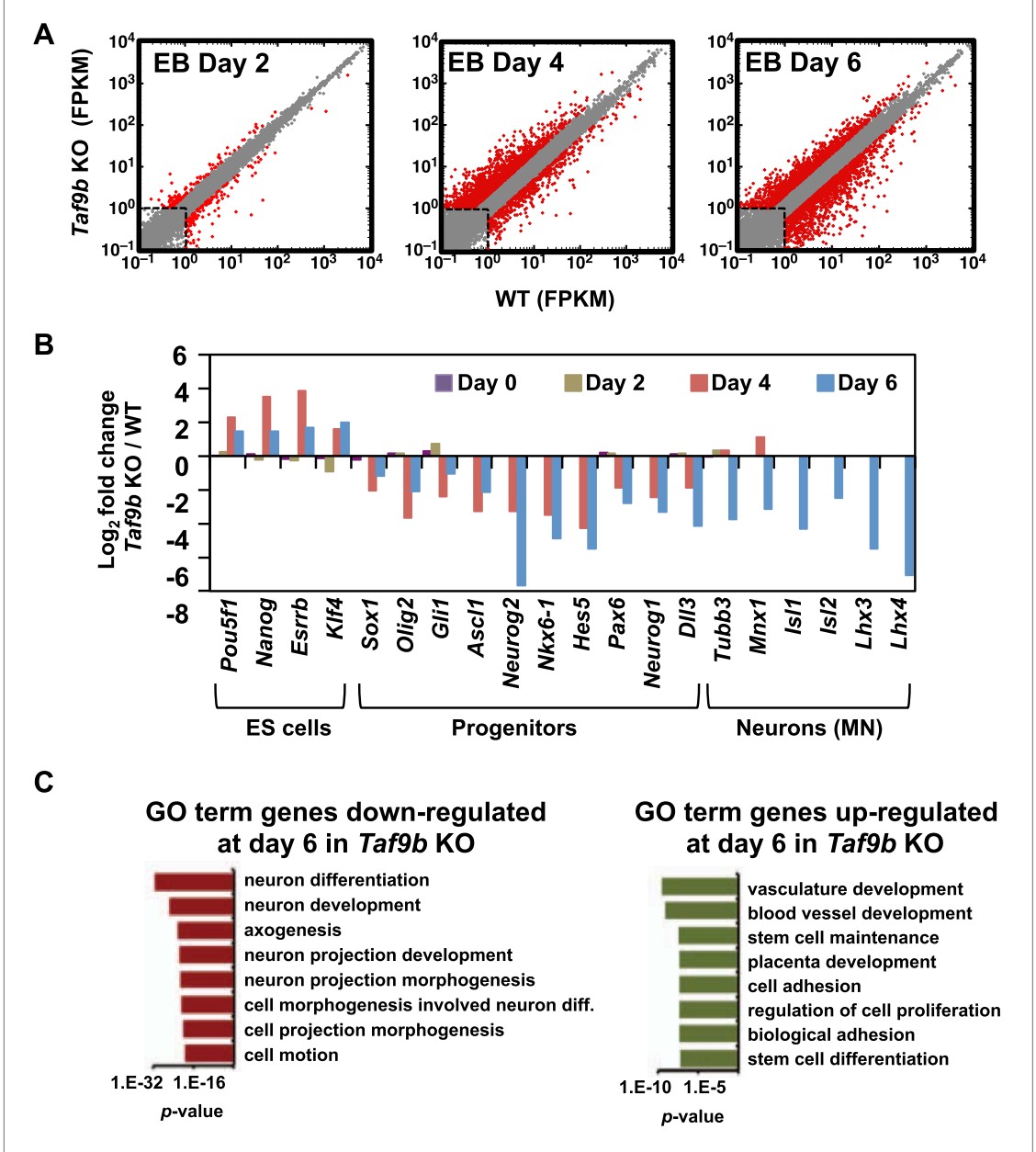

**Figure 4**. TAF9B is specifically required for the efficient activation of neuronal gene expression during in vitro motor neuron differentiation. (**A**) RNA-seq analysis of WT and *Taf9b* KO cells at different time points during EB-MN differentiation. Scatter plots represent FPKM values for genes expressed in WT and *Taf9b* KO cells. Red dots are genes whose expression is affected more than twofold with p-values <0.05. Gray dots represent genes not changing more than the selected cutoff. (**B**) Selected genes from RNA-seq analysis are shown as log$_2$ fold change between *Taf9b* KO and WT cells. Examples are given for key genes expressed in ES cells, progenitor cells, and motor neurons. (**C**) GO term analysis of genes affected by the loss of TAF9B in RNA-seq analysis of EB-MN differentiated for 6 days. Lists show the top eight GO term Biological Process categories obtained ranked by p-value.

The following figure supplements are available for figure 4:

**Figure supplement 1**. TAF9B is specifically required for the efficient activation of neuronal gene expression during in vitro motor neuron differentiation.

patterning and dorsal spinal cord development (*Figure 5B*). Several of the most down-regulated genes in the absence of TAF9B detected by RNA-seq (e.g., *Neurog2* and *Lhx4*) were associated with TAF9B distal binding sites (*Figure 5—figure supplement 1A,E*). Moreover, genes bearing distal TAF9B peaks (e.g., 500–5000 bp from TSS) showed a higher percentage of genes down-regulated in *Taf9b* KO neuralized EBs than control gene sets (*Figure 5—figure supplement 1D*). Collectively, these

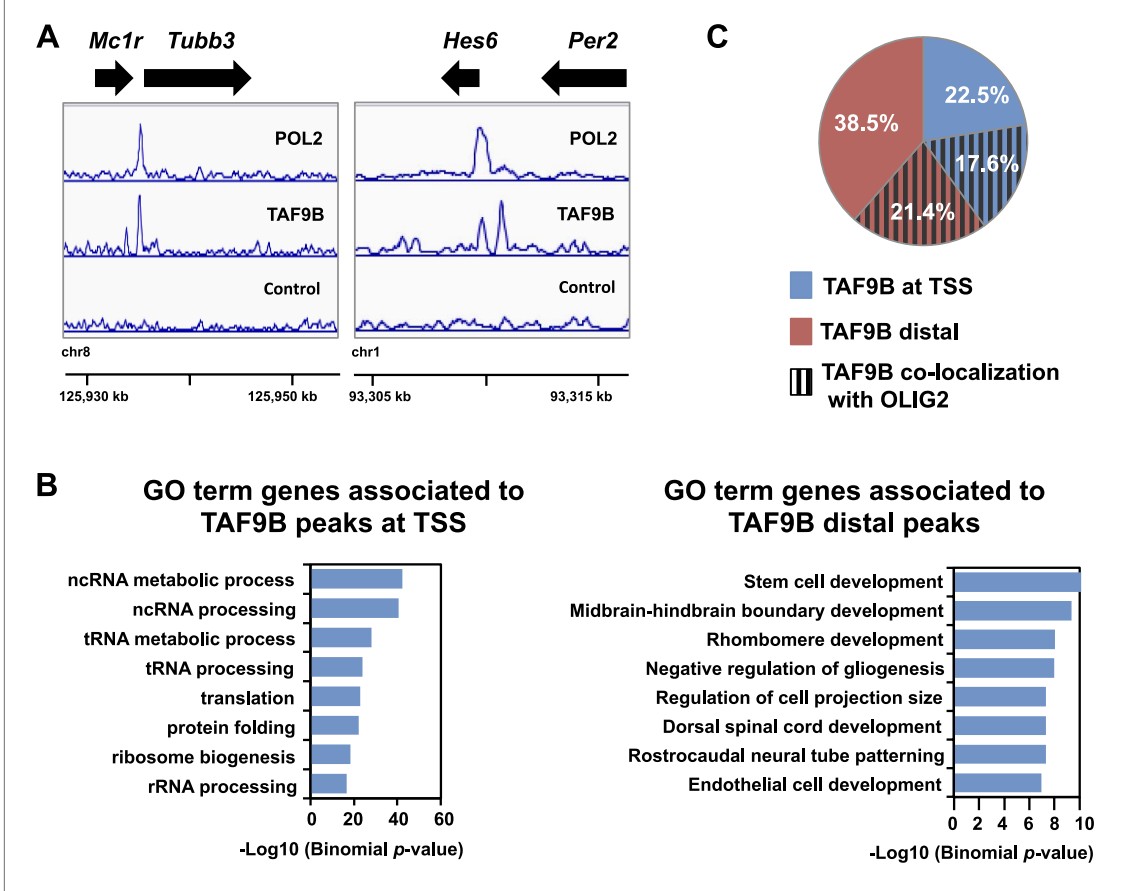

**Figure 5**. TAF9B binds promoter and distal regions of neuronal genes. (**A**) ChIP-seq analysis of TAF9B and RNA POL2 binding sites in EB-MN samples differentiated for 8 days. TAF9B and RNA POL2 ChIP-seq peaks examples are given for *Tubb3* and *Hes6* genes. (**B**) List of top GO terms Biological Process of genes associated to TAF9B distal peaks and TAF9B peaks at annotated transcription start sites (TSS) ranked by p-value. Association of ChIP-seq peak to annotated genes and GO analysis was performed using GREAT. (**C**) Co-localization of TAF9B distal peaks and TAF9B peaks at TSS with OLIG2 ChIP-seq peaks in EB-MN samples.

The following figure supplements are available for figure 5:

**Figure supplement 1**. TAF9B binds promoter and distal regions of neuronal genes.

results suggest that distal TAF9B binding sites are likely functional regulatory elements controlling neuronal genes. We subsequently compared the binding sites of TAF9B to the binding sites of OLIG2, a key transcriptional activator involved in motor neuron differentiation mapped previously (*Mazzoni et al., 2011*). We found co-localization between TAF9B and OLIG2 at both the TAF9B distal peaks and at the TAF9B TSS peaks, suggesting that TAF9B and OLIG2 act in concert at a subset of neuronal genes to regulate gene expression during motor neuron development (*Figure 5C*). The incomplete overlap of TAF9B and OLIG2 peaks may in part represent intrinsic differences between progenitor cells and post-mitotic neurons, since the OLIG2 binding was performed with neuralized EBs enriched for progenitor cells while TAF9B mapping was done with EBs enriched for post-mitotic motor neurons. In addition, it is likely that TAF9B co-localizes with activators other than OLIG2 in this developmental pathway.

## TAF9B is associated with a PCAF-containing complex

Some TAFs are known to be part of not only the TFIID complex but also the SAGA complex (Spt-Ada-Gcn5-acetyltransferase), which can mediate histone acetylation and deubiquitination (reviewed by *Spedale et al., 2012*). In human HeLa cells, TAF9B had previously been reported to be a component of both the TFIID and SAGA complexes (*Frontini et al., 2005*). To test whether TAF9B in mouse neurons is primarily associated with TFIID or SAGA-like complexes, we performed co-immunoprecipitation (co-IP)

experiments with antibodies against TAF9B using in vitro differentiated motor neurons. We observed little detectable association of TAF9B with TAF4 and TAF7, core components of the TFIID complex, while TBP antibodies co-IP'ed those TFIID components efficiently. In contrast, we observed an association between TAF9B and PCAF, a histone acetyltransferase present in the mammalian SAGA complex (*Figure 6A*). Immunoprecipitations of 293T cells transfected with flag-tagged versions of TAF9B or TAF9 confirmed that these IPs are specific for TAF9B and not the closely related TAF9 subunit (*Figure 6B*). We also performed TAF9B co-IP experiments on dissected spinal column tissue samples from WT and *Taf9b* KO newborn mice, confirming that the interactions between TAF9B and PCAF occur in vivo (*Figure 6C*). Thus, TAF9B primarily associates with PCAF likely in a SAGA-like complex rather than the canonical TFIID in murine neurons.

## TAF9B controls neuronal gene expression in vivo

To test the in vivo role of TAF9B during neuronal development, we generated a KO mouse using the *Taf9b* KO ES cells obtained from KOMP (*Figure 7—figure supplement 1A*). *Taf9b* KO mice were viable and fertile, though the number of pups and birth weights were reduced in *Taf9b* KO matings compared to WT controls (*Figure 7—figure supplement 1C*). As expected, spinal cord isolated from newborn KO animals did not express TAF9B (*Figure 7A,B*, *Figure 7—figure supplement 1D*). Given that *Taf9b* KO cells in vitro showed clear de-regulation of genes in both progenitor cells and post-mitotic motor neurons (*Figure 4*), we surmised that loss of TAF9B in vivo may affect not only motor neuron formation but perhaps also proper differentiation of other cell types present in the spinal cord. To test whether neuronal gene expression in vivo is specifically affected in the absence of TAF9B, we carried out RNA-seq analysis to compare whole lumbar spinal column tissue dissected from newborn WT and KO littermates (*Figure 7C*). Our data indicate that genes down-regulated in the absence of TAF9B are enriched for neuronal genes including gene categories such as neuron projection, synapse and axonogenesis (*Figure 7D*). Among the down-regulated genes identified by RNA-seq were neuronal genes such as *Map1b*, *L1cam*, *Nefm*, *Nefh*, and *Isl2*. Gene expression analysis by qRT-PCR of lumbar spinal column tissues confirmed that those neuronal genes are consistently down-regulated in KO compared to WT littermate controls (*Figure 7E*). Several different markers of specific neuronal populations, as well as the general marker *Tubb3*, were also affected in the absence of TAF9B, suggesting a global defect in neuronal gene expression in the spinal cord (*Figure 7—figure supplement 1E*; *Supplementary file 2*). These results are also consistent with a global reduction in neuronal tissue compared to surrounding non-neuronal tissue in the dissected spinal column preparations. Our data suggest that the observed pan-neuronal down-regulation and loss of neuronal tissue may be due to problems in the control of gene expression that coordinates the precise differentiation of progenitor cells during neuronal development. These in vivo results taken together with our in vitro differentiation studies strongly support the notion that TAF9B is specifically required for the regulation of neuronal genes during neuronal development in the spinal cord.

## Discussion

The control of tissue-specific gene expression has been largely attributed to the combinatorial action of transcriptional activators and repressors. These regulatory factors are thought to interact in concert with a highly conserved and prototypic set of core promoter recognition proteins and co-activators that form the PIC at

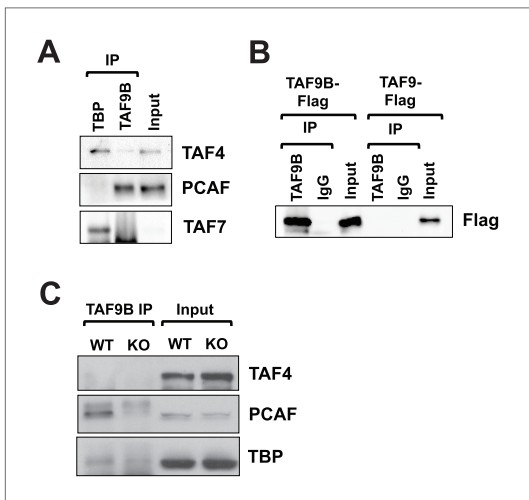

**Figure 6**. TAF9B is associated with PCAF. (**A**) TAF9B and TBP were immunoprecipitated from EB-MN cells differentiated for 8 days and co-immunoprecipitated proteins analyzed by western blotting using antibodies against the TFIID subunits TAF4 and TAF7, and the histone acetyltransferase PCAF. (**B**) 293T cells were transfected with pCMV-3xFLAG-TAF9B or -TAF9, immunoprecipitated with TAF9B antibodies, and analyzed by western blotting using anti-Flag antibodies. (**C**) TAF9B was immunoprecipitated from spinal column extracts from WT and *Taf9b* KO newborn mice and co-immunoprecipitated proteins were analyzed by western blotting detecting TAF4, TBP and PCAF.

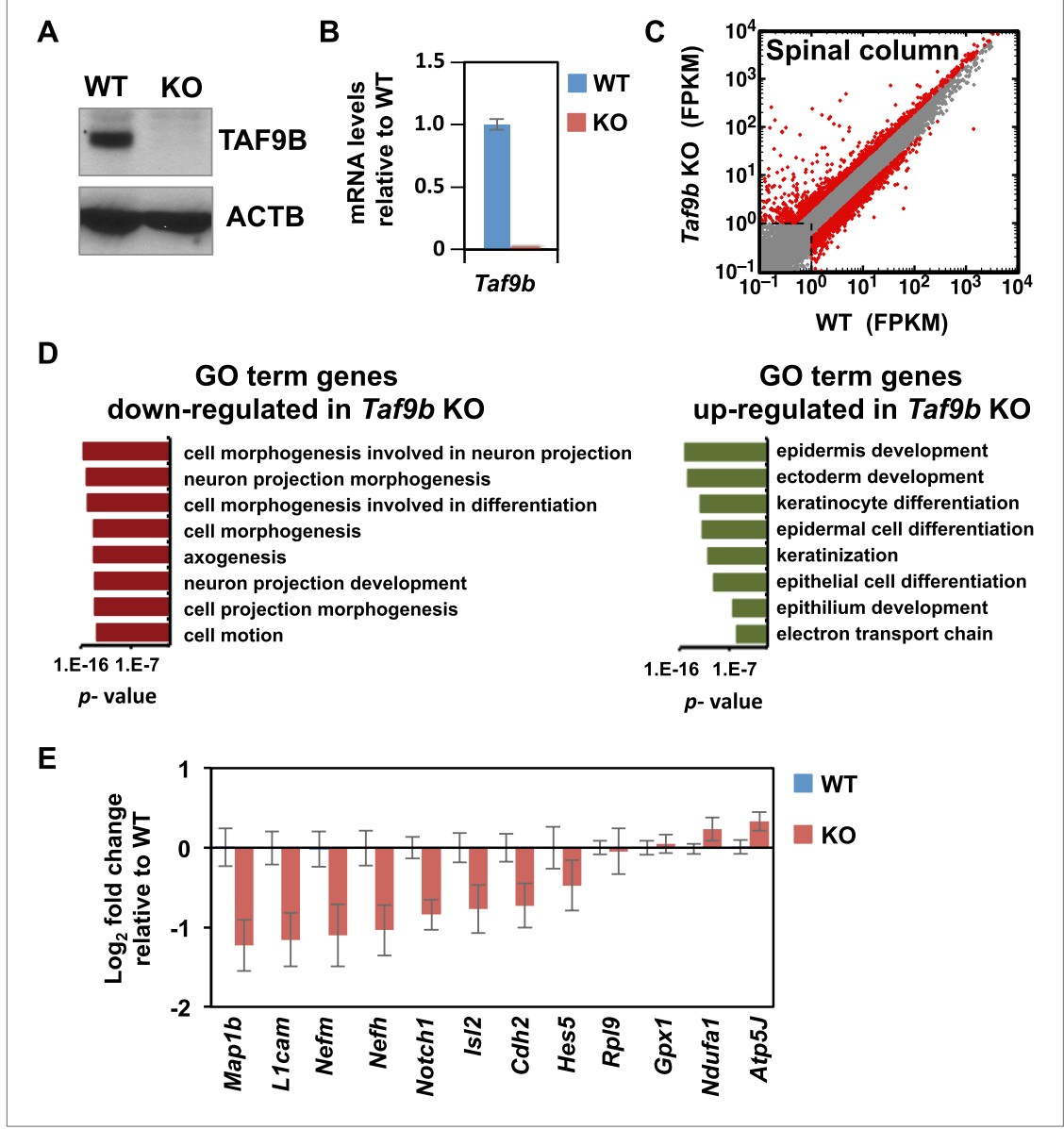

**Figure 7**. TAF9B controls neuronal gene expression in vivo. (**A**) Western blot analysis of spinal cord tissue from WT and *Taf9b* KO newborn mice detecting TAF9B and ACTB as control. (**B**) Spinal cord tissue from WT and *Taf9b* KO newborn mice were dissected and analyzed by qRT-PCR for *Taf9b* expression. Graph shows mean ± SEM of three biological replicates. (**C**) RNA-seq analysis of dissected lumbar spinal column tissue from WT and *Taf9b* KO newborn mice. Scatter plot represents FPKM values of genes expressed in WT and *Taf9b* KO samples. Red dots are genes whose expression is affected more than twofold with p-values <0.05. Gray dots represent genes not changing more than the selected cutoff. (**D**) GO term analysis of genes affected by the loss of TAF9B in RNA-seq analysis. List shows the top eight GO term Biological Process categories obtained ranked by p-value. (**E**) Gene expression analysis by qRT-PCR of lumbar spinal columns tissues of newborn mice. Graphs represent mean ± SEM of littermate comparisons (WT n = 6, KO n = 8).

The following figure supplements are available for figure 7:

**Figure supplement 1**. TAF9B controls neuronal gene expression in vivo.

gene promoters. However, this model has been challenged by different studies demonstrating changes that occur in the core promoter machinery in a tissue-specific manner (reviewed by *D'Alessio et al., 2009*; *Goodrich and Tjian, 2010*). Despite the emerging roles that these core promoter components play in regulating cell-type specific gene expression, relatively little was known about possible changes that may occur in the composition of core promoter factors during neuronal differentiation. In this

study, we screened for changes in the expression of core promoter factors including TAFs, general transcript factors (GTFs), and mediator subunits (MEDs) upon neuronal differentiation. We found that most of the components in the mammalian PIC are expressed at relatively low levels in dissected spinal cord tissue compared to ES cells. Likewise, we observed down-regulation of several of these core promoter recognition components during in vitro differentiation of murine ES cells into motor neurons. By contrast, one orphan TAF, TAF9B, becomes highly up-regulated during motor neuron formation. Using a loss-of-function approach, we found that TAF9B is required for the efficient expression of neuronal genes upon in vitro motor neuron differentiation. Importantly, we found that TAF9B is dispensable for global gene expression in undifferentiated ES cells in contrast to the critical role that other TAFs in TFIID play in these cells (*Pijnappel et al., 2013*). Using ChIP-seq, we found that TAF9B binds promoter and distal regulatory regions of neuronal genes co-localizing in a subset of regions with OLIG2, a key regulator of motor neuron differentiation. Strikingly, *Taf9b* KO mice showed defects in the expression of neuronal genes examined by transcriptome analysis of whole spinal column tissues. Importantly, the mechanism by which TAF9B mediates neuronal-specific transcription programs is unlike all previous TAF involvement in cell specific function: rather than operating through TFIID or distal enhancer complexes (*Freiman et al., 2001*; *Liu et al., 2011*; *Zhou et al., 2013b*), it associates with a SAGA/PCAF co-activator complex.

Our results and those of several other studies probing tissue-specific activities of core promoter factors and co-activators suggest a model in which changes in these complexes are integral to the transcriptional program leading to terminal differentiation. Different cell types employ distinct combinations of activators that must engage in specific protein–protein interactions with core promoter factors. We speculate that rather than existing as a static complex awaiting recruitment by these transcription factors, core promoter complexes themselves must undergo compositional changes using cell type-specific components such as TAF9B. These alternative promoter complexes expand the diversity and number of possible protein–protein transactions necessary to achieve proper fine-tuned control of gene expression. Furthermore, we speculate that such diversified core machineries might provide a mechanism to help lock down and maintain the functions and identity of differentiated cell types. We imagine that in a given cell type, if the silencing of an undesirable activator or repressor is imperfect, their specific interacting partners in the targeted core machinery may nevertheless not be present, thus serving as a fail safe mechanism to prevent unscheduled alterations in programs of gene transcription. We suspect that such core promoter complex dependent control and maintenance of cell identity would likely be accompanied by activator selectivity and integrated with changes in chromatin structure and other layers of regulation that must work coordinately to keep expression patterns stable yet responsive to different stimuli in terminally differentiated cells. We anticipate that further analysis of adult neuronal cell-types as well as other differentiated somatic cells will bring new insights regarding the use of alternative core machineries that regulate gene expression in normal and disease states.

To start addressing these questions in the context of tissues in vivo, we generated *Taf9b* KO mice. We found that mice lacking TAF9B are viable, indicating that this orphan TAF is either not critical for normal development of cell-types other than neurons or its function is partly redundant with the closely related TAF9 homolog. We established that indeed *Taf9* is highly expressed in both mouse ES cells and neurons (as detected by RNA-seq). It is possible that TAF9 partially compensates for the loss of TAF9B thus blunting further global gene expression defects in *Taf9b* KO cells. In human HeLa and chicken DT40 cells partially redundant functions have been observed for these paralogs (*Chen and Manley, 2003*; *Frontini et al., 2005*). It will be interesting to test whether *Taf9* KO mouse are viable and whether the double KO of these two related genes (*Taf9* and *Taf9b*) results in more pronounced phenotypes. No *Taf9* KO mice have been reported to date. *Taf4b* and *Taf7l* KO mice are also viable, but unlike *Taf9b* KO mice, they are infertile (*Freiman et al., 2001*; *Falender, 2005*; *Zhou et al., 2013a*). Our results suggest that TAF9B is not critical for gametogenesis, in contrast to what has been observed for these two other orphan TAFs. Instead, our results point to a model where TAF9B is involved in the activation of neuronal genes by binding to distal and promoter proximal DNA regulatory elements associated with the histone acetyltransferase PCAF that, like its paralog GCN5, are subunits of the SAGA/STAGA/TFTC complex. This complex contains several TAFs shared with TFIID including TAF6, TAF9, TAF5, TAF10, TAF12, and a de-ubiquitinase (DUB) module that removes ubiquitin from histone H2B (reviewed by *Spedale et al., 2012*; *Weake and Workman, 2012*). Because TAFs are structural components of both SAGA and TFIID complexes, it is possible that TAF9B is affecting the competitive recruitment or activity of a SAGA-like complex in place of TFIID.

Among the genes affected by the loss of TAF9B are several targets of the Notch- and Shh-signaling pathway, both known to play key roles in neuronal development (reviewed by *Louvi and Artavanis-Tsakonas, 2006*; *Jessell, 2000*). Interestingly, several lines of evidence connect the activation of Notch dependent genes with the PCAF-containing complex providing a potential link between PCAF, TAF9B, and activation of these genes. For example, PCAF has been shown to cooperate with another co-activator protein, p300, in Notch intracellular domain-dependent transcription assays using chromatin templates in vitro, as well as in transfection/reporter assays (*Kurooka and Honjo, 2000*; *Wallberg et al., 2002*). Other components of the SAGA complex have been linked to the specific control of gene expression in cells of the central nervous system. Mutations in two subunits of the *Drosophila* DUB complex, Nonstop and Sgf11, lead to defects in neural development in the optic lobe (*Weake et al., 2008*). In mice, mutations in the HAT domain of GCN5 result in defects in cranial neural tube closure and embryonic lethality (*Bu et al., 2007*). In humans, mutations in the DUB subunit ATXN7 are associated with the development of spinocerebellar ataxia type 7, a disease characterized by the loss of motor control as well as retinal defects (reviewed by *Koutelou et al., 2010*). Interestingly, we observed that *Taf9b* KO mice display defects in eye development including microphthalmia and cataracts-like phenotypes albeit with modest penetrance (data not shown). The type of neuronal-specific role performed by TAF9B is reminiscent of the cell-type specific roles of particular subunits of the ATP-dependent chromatin remodeling complex BAF. This complex undergoes changes in composition upon neuronal differentiation as specific BAF subunits are incorporated in neuronal progenitors cells as well as in post-mitotic neurons (*Olave et al., 2002*; *Lessard et al., 2007*).

In this study, we have focused on characterizing the molecular phenotypes and have documented defects in gene regulation due to the absence of TAF9B during in vitro differentiation of motor neurons and in vivo in spinal column tissues. It is also possible that the down-regulation of neuronal genes observed in whole spinal column tissues in *Taf9b* KO mice may represent a specific loss of neuronal tissue relative to the surrounding vertebrae due perhaps to defects in neuronal progenitor cells during embryonic development. In future experiments, we hope to address the extent of potential neurological defects in *Taf9b* KO mice, including deficits in motor skills as well as other potential neurological defects due to the role that TAF9B may be playing in other neuronal populations besides motor neurons. Preliminary experiments using *Taf9b* KO mice carrying the *LacZ* reporter gene suggest that *Taf9b* is highly expressed in the developing hypothalamus at mid-gestation (data now shown). In a recent study describing neuronal activity-dependent ribosome profiling in the adult mouse hypothalamus (*Knight et al., 2012*), *Taf9b* can be identified in the genomic data set as highly expressed in this tissue compared to several other TAFs. Moreover, the levels of *Taf9b* mRNA associated with ribosomes increased when neurons in the hypothalamus were activated, while several other TAFs were expressed at low levels or not enriched in a neuronal activity-dependent manner. These results suggest that TAF9B may be involved in setting up transcriptional responses in at least certain neuronal circuits in the hypothalamus. In the future, it will be interesting to directly test the role of TAF9B in the adult hypothalamus and to extend the analysis to other adult neuronal types.

## Materials and methods

### Cell culture

Murine embryonic stem cells were grown in embryoMAX DMEM (EMD Millipore, Billerica, MA) supplemented with LIF and 15% FBS. Cells were differentiated into embryoid bodies enriched in motor neurons (EB-MN) as described previously (*Wichterle et al., 2002*). $2 \times 10^5$ cells/ml were incubated in differentiation media (25% embryoMAX DMEM, 25% F12 media, 25% neurobasal media, 1x B27 supplement, 1.5 mM L-glutamine and 0.1 mM β-mercaptoethanol) supplemented with 1 μM retinoic acid and 0.8 μM smoothened agonist SAG (N-Methyl-N′-[3-pyridinylbenzyl]-N′-[3-chlorobenzo{b}thiophene-2-carbonyl]-1,4-diaminocyclohexane, VWR, Radnor, PA) for up to 8 days. Samples were collected at different times as indicated. SHH conditioned media was generated by transfecting 293T cells with a vector expressing Shh-N and used when noted. The Notch signaling pathway inhibitor DAPT (N-[N-{3,5-difluorophen- acetyl}-l-alanyl]-S-phenylglycine t-butyl ester, Tocris Bioscience, United Kingdom) was added when noted to deplete progenitor cells and enrich for post-mitotic neurons in the culture (*Crawford and Roelink, 2007*). The *Taf9b* KO mouse ES cells (Taf9b$^{tm1(KOMP)Vlcg}$) were generated by the trans-NIH Knock-Out Mouse Project (KOMP) and obtained from the KOMP Repository (www.komp.org). NIH grants to Velocigene at Regeneron Inc (U01HG004085) and the CSD Consortium (U01HG004080)

funded the generation of gene-targeted ES cells for 8500 genes in the KOMP Program and archived and distributed by the KOMP Repository at UC Davis and CHORI (U42RR024244). We confirmed the described deletion by DNA sequencing (data not shown). Since the *Taf9b* gene is located in the X chromosome this male KO ES cells are a complete null for this gene. JM8 WT murine ES cells are from the same genetic background as *Taf9b* KO ES cells (C57BL/6). JM8 and *Taf9b* KO ES cells were grown on MEF feeder layers inactivated with mitomycin C (Sigma-Aldrich, St. Louis, MO). Murine ES cells carrying a GFP reporter under *Mnx1* promoter (HBG3) were described previously (*Wichterle et al., 2002*).

## qRT-PCR

Total RNA was isolated using RNeasy kit (Qiagen, Germany) and cDNA was generated using Superscript III RT system using manufacture's instructions (Life Technologies, Carlsbad, CA). Primers used in these assays are described in *Supplementary file 3*. Relative expression levels in in vitro time course differentiation experiments were normalized to total RNA. Relative levels of expression of spinal column tissues were normalized to *Gapdh*. PCR reactions were performed using SYBR Green PCR Master Mix according to the manufacturer's instructions in an ABI 7300 real time PCR machine (Applied Biosystems, Grand Island, NY).

## Vector, plasmids and transfections

*Taf9b* and *Taf9* cDNAs were obtained from Open Biosystems clones (TAF9B clone ID#30468965, TAF9 clone ID#5006430) and cloned into p3xFLAG-CMV-10 vector to obtain p3xFLAG-CMV-TAF9B and -TAF9. Plasmids were transfected into 293T cells by lipofectamine 2000 (Life Technologies). Cells were collected 48 hr after transfection.

## Co-Immunoprecipitations

Transfected 293T cells were lysed and the protein concentration was measured by the Bradford method. EB-MN differentiated for 8 days were collected and nuclear extract was prepared as previously described (*Pugh, 1995*). Spinal column tissues were dissected from newborn mice, immediately frozen in liquid nitrogen, and stored at −80°C. Tissue samples were ground into powder in liquid nitrogen and cell lysis buffer was added to extract proteins. After centrifugation, the protein concentration was measured by the Bradford method. Indicated antibodies were mixed with protein A sepharose (GE Healthcare life science, Pittsburgh, PA) or protein G Dynabeads (Life technologies) for 1 hr at 4°C. Protein extracts (1 mg) were mixed with the antibodies/beads complexes for overnight at 4°C on a rotating wheel. The beads were washed three times with washing buffer containing 300 mM KCl and 0.05% NP-40 and once with washing buffer containing 100 mM KCl. Samples were boiled in SDS loading buffer for 5 min and analyzed by Western blotting using indicated antibodies.

## Western blots and immunostainings

Western blots were performed using whole cell extracts and SDS-PAGE. Immunostainings of EB-MN were performed after fixing samples in 4% paraformaldehyde for 30 min at room temperature. Antibodies were incubated in PBS, 0.1% Triton X-100 and washed three times with PBS, 0.1% Triton X-100 for 10 min each time.

## Antibodies

Antibodies used are anti-TBP (cat#51841; Abcam, Cambridge, Massachusetts), anti-TAF4 (cat#612054; BD Biosciences, San Jose, CA), anti-TAF7 (cat#H00006879-M01; Abnova, Taiwan), anti-TAF9B (cat#A303-810A; #G2306 and Bethyl Laboratories, Montgomery, TX), anti-TAF10 (Santa Cruz cat#102125), anti-OCT4 (Santa Cruz cat#8628, Dallas, Texas), anti-TUBB3 (cat#MMS435P; Covance, Princeton, New Jersey), anti-ACTB (cat#A2228; Sigma-Aldrich), anti-RNA POL2 (8WG16; Clone), anti-PCAF (cat#13124; Santa Cruz), anti-ISL1/2 (Gift from the Jessell Lab, Columbia University), anti-NANOG (cat#A300-397A; Bethyl Laboratories), anti-Ki67 (cat#16667; Abcam), anti-FLAG (cat#F3165; Sigma-Aldrich) and Rabbit IgG control (cat#46540; Abcam). TAF9B antibodies #G2306 were produced by injecting a peptide corresponding to residues 226–245 of TAF9B into rabbit by OpenBiosystems (GE Healthcare). Antiserum was tested for specificity using extracts from 293T cells transfected with p3xFLAG-CMV-TAF9B and -TAF9.

## Cell cycle and apoptosis assays

Cell cycle was measured using BrdU Flow kit following manufacturer's instructions (cat#552598; BD Biosciences). Apoptosis was measured using PE annexin V apoptosis detection kit following manufacturer's instructions (cat#559763; BD Biosciences).

## RNA-seq

Total RNA was isolated from cells or tissue extracts using RNeasy Kit (Qiagen) and 1.5 µg of total RNA was used to prepared Poly-A RNA-seq libraries following Illumina protocols. Samples were sequenced using Illumina Sequencers at the QB3 Vincent J Coates Genomics Sequencing Laboratory, UC Berkeley. Sequenced reads were mapped against RefSeq genes using TopHat (*Trapnell et al., 2009*) and differentially expressed genes were determined using Cuffdiff (*Trapnell et al., 2010*). Genes expressed more than 1 FPKM and with >twofold difference and p-value < 0.05 as calculated by Cuffdiff were considered as differentially expressed for scatter plots representations and for GO term analysis using DAVID Bioinformatic resources (*Huang et al., 2009*).

## Chromatin IP and ChIP-seq

ChIP-seq was performed using 1 mg of chromatin for each IP reaction using antibodies against TAF9B (#G2306), RNA POL2 (8WG16), and IgG control with EB-MN differentiated for 8 days. EB-MN samples were fixed with 0.5% formaldehyde for 10 min at room temperature and ChIPs were performed as described previously (*Liu et al., 2011*; *Zhou et al., 2013b*). ChIP-qPCR experiments were performed using EB-MN samples differentiated for 6–8 days. ChIP-seq libraries were prepared following Illumina protocols and sequenced at the QB3 Vincent J Coates Genomics Sequencing Laboratory, UC Berkeley. Reads were mapped to UCSC version mm9 using Bowtie (*Langmead et al., 2009*). ChIP-seq peaks were determined using MACS (*Zhang et al., 2008*) and significant peaks were associated to genes using GREAT default parameters (*McLean et al., 2010*).

## Mouse *Taf9b* KO

All animal experiments were performed in strict accordance with the recommendations in the guide for the care and use of laboratory animals of the National Institutes of Health and following the animal use protocol (#R007) approved by the Animal Care and Use Committee (ACUC) of the University of California, Berkeley. *Taf9b* KO mouse was generated in the UC Berkeley transgenic facility by injecting *Taf9b* KO murine ES cells (TAF9b[tm1(KOMP)Vlcg]) into albino C57BL/6 background (C57BL/6J-*Tyr*[c-2J]) and chimeric mice were obtained. *Taf9b* heterozygous animals were backcrossed to WT C57BL/6 mice to obtain *Taf9b* heterozygous mice without the albino phenotype (*Tyr*[c-2J]). Mice were genotyped by PCR using primers detecting WT and KO sequences. Primers sequences are available in *Supplementary file 3*.

## Acknowledgements

The authors would like to thank Gina Dailey, Shuang Zheng, and Patrick Visperas for technical assistance, Tom Jessell (Columbia University) for antibodies and discussion, and Claudia Cattoglio and Ivan Grubisic for assistance in the analysis of genomic data. Sharon Torigoe, Sheila Teves, Jaclyn Ho, and Claudia Cattoglio for critical reading of the manuscript. Benjamin Guglielmi and members of the Tjian lab for critical discussions. FH was a CIRM Postdoctoral Scholar of the UC Berkeley Stem Cell Center and currently a Research Specialist of the Howard Hughes Medical Institute. TY was a Jane Coffin Childs Fund fellow. HR is supported by 5R01GM097035. RT is an Investigator of the Howard Hughes Medical Institute.

## Additional information

### Competing interests

RT: Robert Tjian is President of the Howard Hughes Medical Institute (2009-present), one of the three founding funders of *eLife*. The other authors declare that no competing interests exist.

### Funding

| Funder | Grant reference | Author |
| --- | --- | --- |
| Howard Hughes Medical Institute (HHMI) | 003052 | Francisco J Herrera, Teppei Yamaguchi, Robert Tjian |
| National Institutes of Health (NIH) | CA25417 | Francisco J Herrera, Teppei Yamaguchi, Robert Tjian |
| California Institute for Regenerative Medicine (CIRM) | RT1-01021 | Francisco J Herrera, Teppei Yamaguchi, Robert Tjian |

| Funder | Grant reference | Author |
|---|---|---|
| California Institute for Regenerative Medicine (CIRM) | T1-00007 | Francisco J Herrera |
| Jane Coffin Childs Memorial Fund for Medical Research | 61-1439 | Teppei Yamaguchi |
| National Institutes of Health (NIH) | 5R01GM097035 | Henk Roelink |
| California Institute for Regenerative Medicine (CIRM) | RB4-06016 | Teppei Yamaguchi, Robert Tijan |

The funders had no role in study design, data collection and interpretation, or the decision to submit the work for publication.

### Author contributions
FJH, Conception and design, Acquisition of data, Analysis and interpretation of data, Drafting or revising the article; TY, Acquisition of data, Analysis and interpretation of data, Drafting or revising the article; HR, Analysis and interpretation of data, Drafting or revising the article; RT, Conception and design, Analysis and interpretation of data, Drafting or revising the article

### Ethics
Animal experimentation: All animal experiments were performed in strict accordance with the recommendations in the guide for the care and use of laboratory animals of the National Institutes of Health and following the animal use protocol (#R007) approved by the Animal Care and Use Committee (ACUC) of the University of California, Berkeley.

## Additional files

### Supplementary files
• Supplementary file 1. GO Term annotation of RNA-seq and ChIP-seq results.

• Supplementary file 2. Gene expression analysis of neuronal markers in *Taf9b* KO mouse.

• Supplementary file 3. Sequences of oligonucleotides used in this study.

### Major datasets
The following dataset was generated:

| Author(s) | Year | Dataset title | Dataset ID and/or URL | Database, license, and accessibility information |
|---|---|---|---|---|
| Herrera FJ, Yamaguchi T, Roelink H, Tjian R | 2014 | Core promoter factor TAF9B regulates neuronal gene expression | http://www.ncbi.nlm.nih.gov/geo/query/acc.cgi?acc=GSE55782 | Publicly available at NCBI Gene Expression Omnibus. |

The following previously published dataset was used:

| Author(s) | Year | Dataset title | Dataset ID and/or URL | Database, license, and accessibility information |
|---|---|---|---|---|
| Mazzoni EO, Mahony S, Iacovino M, Morrison CA, Mountoufaris G, Closser M, Whyte WA, Young RA, Kyba M, Gifford DK, Wichterle H | 2011 | Embryonic stem cell based system for the discovery and mapping of developmental transcriptional programs | http://www.ncbi.nlm.nih.gov/geo/query/acc.cgi?acc=GSE30882 | Publicly available at NCBI Gene Expression Omnibus. |

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
