## [Decision Letter]

Thank you for sending your work entitled “Core promoter factor TAF9B regulates neuronal gene expression” for consideration at *eLife*. Your article has been favorably evaluated by a Senior editor and 3 reviewers, one of whom, Robb Krumlauf, is a member of our Board of Reviewing Editors.

The Reviewing editor and the other reviewers discussed their comments before we reached this decision, and the Reviewing editor has assembled the following comments to help you prepare a revised submission.

It is the composite view of the reviewers that the work by Herrera et al. is in line with the mounting evidence that some of the so called “general transcription factors” regulate cell specific gene expression in addition to their roles in general transcription. The authors report here that the TBP-related factor TAF9B is strongly and selectively induced upon motor neuron differentiation, and is required for the expression of a subset of neuronal-lineage genes. Taf9b mutant ESCs do not show a gross perturbation in their gene expression profile but neuronal gene marker transcription is selectively disrupted and their ability to be differentiated into MN is compromised. In agreement with the central hypothesis of this work, ChIP-seq analysis reveals that Taf9b associates with not only proximal elements but also to distal presumptive regulatory regions that are engaged by the sequence and lineage specific transcription factor Olig2. Moreover, Taf9b associates poorly with Taf4 and Taf7 but strongly with PCAF. Gene expression analysis of the spinal cord of Taf9b mutant mice reveals an altered expression profile where some genes expressed in neurons are down-regulated. Overall, the identification of TAF9B as a lineage-specific transcriptional regulator important for neuronal differentiation in cell culture and mice is an important contribution. There will clearly be strong interest in TAF9B going forward as a modulator/regulator of ESC differentiation and potentially also in hypothalamus function, for which this paper will be an important resource.

There are, however, some concerns that need to be addressed. One major issue is that the central message of the manuscript is stated as the “Core promoter factor TAF9B regulates neuronal gene expression”. The authors clearly demonstrate that in RA/Shh induced MN differentiation in ESCs is impaired. Figures 3 and 4 suggest that ESCs do not transition well into early stages of neuronal differentiation (i.e., lack of Sox1 and Pax6 induction and retention of Oct4 expression). However, this general neuronal differentiation phenotype conflicts slightly with the mutant phenotype in mice. Based on the data presented in Figure 7, rather than affecting general neuronal differentiation it is equally plausible that Taf9b is required for the differentiation of some specific neuronal fates during spinal cord development. To formally test and explore this idea the authors could easily take advantage of the existing RNA-seq data for control and mutant spinal cord tissue to examine the expression of several known genes expressed in different neuronal populations of the spinal cord (PMID:21729788). They might even consider following that with expression studies in spinal cord sections of some of these markers. Considering this possibility will not only consolidate the message, but it will make the case even stronger: the “general transcription factor” Taf9b is required for the differentiation of specific neuronal types.

Minor points to consider are:

1) Specific gene targets strongly regulated by Taf9B are identified in this study (Mnx1, Lhx3, Lhx4, Isl1, Tubb3). However only very limited ChIP-Seq data is provided for specific genes (Ccnc and Calm1), which are distinct from those shown to be dependent on TAF9B. Please show the occupancy of TAF9B on the former set of neuronal genes, and also include markers to indicate the size of the genomic region and genes. Is transcription of Ccnc and Calm1 dependent on TAF9B?

2) Occupancy of TAF9B is compared with that of RNAPII (POL2), but these are active genes so why is POL2 predominantly only at the TSS rather than distributed through the gene?

3) In the graph at the far right in Figure 5, the fragment sizes used are different lengths (i.e., 0 to 5kB versus 50-500kB), and makes it appear that TAF9B is spread out everywhere along the gene with relatively equal frequency, whereas this does not appear to be the case in the examples shown. Showing examples from the Mnx1, Lhx3, etc genes should clarify this point.

4) The biochemical evidence that TAF9B associates with PCAF and not TFIID is clear, but there's no functional evidence that the association is relevant. Is PCAF required for the neuronal genes, and/or is there a strong association of TAF9B with PCAF in the ChIP-Seq? The colocalization with Oligo2 is only at 23% of sites, with the majority of TAF9B sites not in association with either POL2 or OLIG2. Presumably the association with PCAF would be stronger. Does PCAF complex levels change upon RA differentiation of the mESC cells to neuronal cell fate? If the association is required, then the PCAF levels may also rise in day 6 differentiation cultures.

5) Due to the possible role of TAF9B on late stages of MN differentiation, and the misregulation of several “patterning” and “axon guidance” genes based on GO terms, it could be informative to compare TAF9B with Hoxc9 binding. Hoxc9 controls patterning and the MN axon guidance in the thoracic spinal cord (PMID:20826310). The authors might consider exploring whether there is a correlation with Hoxc9, which could strengthen the concept of TAF9B binding with sequence-specific transcription factors to control CNS patterning.

---

## [Author Response]

*One major issue is that the central message of the manuscript is stated as the “Core promoter factor TAF9B regulates neuronal gene expression”. The authors clearly demonstrate that in RA/Shh induced MN differentiation in ESCs is impaired.*
Figures 3 and 4
*suggest that ESCs do not transition well into early stages of neuronal differentiation (i.e., lack of Sox1 and Pax6 induction and retention of Oct4 expression). However, this general neuronal differentiation phenotype conflicts slightly with the mutant phenotype in mice. Based on the data presented in*
Figure 7*, rather than affecting general neuronal differentiation it is equally plausible that Taf9b is required for the differentiation of some specific neuronal fates during spinal cord development. To formally test and explore this idea the authors could easily take advantage of the existing RNA-seq data for control and mutant spinal cord tissue to examine the expression of several known genes expressed in different neuronal populations of the spinal cord (PMID:21729788). They might even consider following that with expression studies in spinal cord sections of some of these markers. Considering this possibility will not only consolidate the message, but it will make the case even stronger: the “general transcription factor” Taf9b is required for the differentiation of specific neuronal types*.

The question raised by the reviewers of whether specific neuronal types in the spinal cord are more affected in *Taf9b* KO mice is indeed of strong interest. To address this question, we have analyzed the expression of different neuronal markers in the dissected lumbar spinal column tissue as suggested by the reviewers. We analyzed in our RNA-seq data all the markers for post-mitotic neuronal populations, as described by Alaynick et al*.* 2011, and the results are presented in the new Supplementary file 2. We observed the down-regulation, to various degrees, of most neuronal specific markers without a clear pattern of deregulation specific for any particular neuronal type. We also performed qRT-PCR analysis for selected marker genes using several WT and *Taf9b* KO littermates and observed the down-regulation, to different extents, of most of the markers tested (New Figure 7—figure supplement 1). There is a strong linear correlation between the RNA-seq and qRT-PCR analysis using different littermates (WT: n=6, KO: n=8) for the 27 genes tested, further validating our genomic data and conclusions. These results suggest that the defects in gene expression observed in *Taf9b* KO mice are likely pan-neuronal, affecting most neuronal cell types, rather than specific neuronal populations. We have included a few sentences describing these findings in the Result section. As mentioned in the Discussion section, a global defect in neuronal gene expression may also be consistent with a specific loss of neuronal tissue relative to other non-neuronal tissues in the lumbar spinal column, perhaps due to defects controlling gene expression in the absence of TAF9B during neuronal development. These results are consistent with our *in vitro* MN differentiation experiments showing that neuronal gene expression is clearly affected at neuronal progenitor as well as post-mitotic MN stage (Figure 4). Although our expression data are more consistent with a global defect in neuronal expression in *Taf9b* KO mice, the lack of spatial information for the defects in gene expression in the spinal cord in our experiments does not allow us to completely rule out that certain neuronal types may be more severely affected than others. We think that further experiments that are beyond the scope of this paper will be necessary to unambiguously address this interesting question. However, the inclusion of several gene markers in our qRT-PCR analysis as well as the detailed analysis of the RNA-seq data presented in Supplementary file 2 strengthens the conclusion that neuronal genes are controlled by TAF9B *in vivo*.

Minor points to consider are:

1) Specific gene targets strongly regulated by Taf9B are identified in this study (Mnx1, Lhx3, Lhx4, Isl1, Tubb3). However only very limited ChIP-Seq data is provided for specific genes (Ccnc and Calm1), which are distinct from those shown to be dependent on TAF9B. Please show the occupancy of TAF9B on the former set of neuronal genes, and also include markers to indicate the size of the genomic region and genes. Is transcription of Ccnc and Calm1 dependent on TAF9B?

As requested by the reviewers, we have included new examples of TAF9B ChIP-seq peaks associated with genes whose expression was affected in *Taf9b* KO cells during differentiation into MNs *in vitro*. Examples are now shown in Figure 5 for *Tubb3* and *Hes6* genes. In Figure 5—figure supplement 1, we included ChIP-seq tracks for OLIG2 and HOXC9 available from a previous study (27) for the same genes and two additional examples, *Mnx1* and *Ngn2*, all of which were down-regulated in the absence of TAF9B in our *in vitro* differentiated MN experiments. New ChIP-qPCR data has been included for selected loci in Figure 5—figure supplement 1. The original examples shown in Figure 5 (*Ccnc* and *Calm1*) were selected to show a representative ChIP-seq peak present at the TSS and at a distal location from the TSS, rather than to highlight the genes most affected by the loss of TAF9B. As seen in other ChIP-seq experiments (*Biggin et al*, 2011), not all genes bound by a transcription factor necessarily become down-regulated in the absence of the targeted factor. In particular, *Ccnc* and *Calm1* are only moderately down-regulated in our *in vitro* MN experiments in the absence of TAF9B without passing our 2 fold cutoff. We have replaced those examples with the genes mentioned above to highlight genes for which we observed binding of TAF9B and concomitant down-regulation (> 2 fold) in TAF9B KO cells in *in vitro* MN differentiation as suggested by the reviewers.

2) Occupancy of TAF9B is compared with that of RNAPII (POL2), but these are active genes so why is POL2 predominantly only at the TSS rather than distributed through the gene?

RNA POL2 occupancy measured by ChIP-seq in actively transcribed genes often displays a peak of high occupancy at the transcription star site (TSS) compared to the gene body. This issue has been widely observed in different studies and has been interpreted as RNA POL2 pausing during the transcription process, but the mechanistic interpretation is still debated in the field. See a recent review by Ehrensberger et al. 2013 for further discussion and references therein.

*3) In the graph at the far right in*
Figure 5*, the fragment sizes used are different lengths (i.e., 0 to 5kB versus 50-500kB), and makes it appear that TAF9B is spread out everywhere along the gene with relatively equal frequency, whereas this does not appear to be the case in the examples shown. Showing examples from the Mnx1, Lhx3, etc genes should clarify this point*.

Our TAF9B ChIP-seq data indicate that when TAF9B is bound at the TSS of genes, the signal does not usually spread to the gene body as seen in the examples provided in Figure 5 and Figure 5—figure supplement 1, as well as for the overall data set (data not shown). The graph described in the original Figure 5 shows the frequency of TAF9B peaks binned by their distance to annotated TSS but does not necessarily represent the spread of ChIP-seq signal along the gene body. This graph and GO term analysis indicate that the majority of TAF9B-only peaks are detected in distal regions relative to TSS of genes and those distal peaks tend to be associated with neuronal genes. To simplify the display of the genomic data we have now modified Figure 5 and Figure 5. Rather than focusing on our original classification of TAF9B-only and TAF9B-POL2 peaks, we subdivided TAF9B peaks based on the proximity to TSS (TAF9B-TSS and TAF9B-distal) and obtained essentially the same results as presented in our original figure. We have moved Figure 5 to Figure 5—figure supplement 1 to highlight the frequencies at which TAF9B can be found at distal regions from TSS.

*4) The biochemical evidence that TAF9B associates with PCAF and not TFIID is clear, but there's no functional evidence that the association is relevant. Is PCAF required for the neuronal genes, and/or is there a strong association of TAF9B with PCAF in the ChIP-Seq? The colocalization with Oligo2 is only at 23% of sites, with the majority of TAF9B sites not in association with either POL2 or OLIG2. Presumably the association with PCAF would be stronger. Does PCAF complex levels change upon RA differentiation of the mESC cells to neuronal cell fate? If the association is required, then the PCAF levels may also rise in day 6 differentiation cultures*.

Several TAFs are known to be components of TFIID and SAGA complex. The association we observed between TAF9B and PCAF, a histone acetyl transferase present in the SAGA complex, is particularly interesting because of several lines of evidence described in the discussion and below connect this complex with neuronal gene expression: (i) PCAF has been identified in *in vitro* and transfection assays as required for the induction of target genes of the Notch signaling pathway. Several genes regulated by the Notch signaling pathway (e.g. *Neurog2* and *Hes5*) were among the most down-regulated genes in our *in vitro* MN differentiation assays in the absence of TAF9B (Figure 4). The loss of TAF9B may affect recruitment and/or function of the PCAF/SAGA complex during the activation of those genes. (ii) ATXN7, one of the subunits of the human SAGA complex, is mutated in Spinocerebellar Ataxia 7, a disease characterized by defect in locomotor and retinal defects. (iii) Mutations in subunits of the SAGA complex (DUB submodule) are associated with defects in neural development in the optic lobe in *Drosophila*. These observations add strong support to the hypothesis that some components of this complex play specific functions in the expression of genes in the central nervous system.

Regarding the co-localization of PCAF and TAF9B in ChIP-seq experiments, currently no ChIP-seq data in differentiated MNs samples are available for PCAF or other subunits of the SAGA/PCAF to compare against TAF9B ChIP-seq peaks. It is likely that other subunits of this complex, including PCAF, will have a strong co-localization with TAF9B in ChIP-seq assays as suggested by the reviewers, but more experiments will be necessary to address this issue. As for OLIG2, we observed a total co-localization with TAF9B peaks of close to 40% in our data. The ∼23% of co-localization mentioned by the reviewers is for only one category of TAF9B peaks (TAF9B-only), another ∼16% of TAF9B peaks co-localized with OLIG2 in the TAF9B-POL2 category. In our new Figure 5 showing TAF9B classified as TAF9B at TSS and TAF9B distal peaks, the same total co-localization with OLIG2 is observed. As mentioned in the discussion, the incomplete co-localization may be due in part to differences in the binding of these factors between progenitor cells and post-mitotic cells, as OLIG2 binding sites were mapped on EB enriched in progenitor cells and TAF9B in post-mitotic MNs. It is also possible that TAF9B may co-localize with other transcriptional activators in this developmental pathway.

Regarding the control of the expression of PCAF upon neuronal differentiation, it is known that PCAF is at least present in two independent complexes, SAGA and ATAC, while TAFs (and TAF9B) are only components of the SAGA, but not the ATAC, complex (Spedale G. et al. 2012). Figure 1—figure supplement 1 has been modified to show the differences in the levels of expression of all subunits of SAGA and ATAC complex between mouse ES cells and spinal cord samples. Interestingly, we observed a few SAGA-specific subunits that are up-regulated in the spinal cord tissue, although not to the same extent of the induction of TAF9B. We also observed that some other SAGA-specific subunits are down-regulated, hinting to the possibility of further changes in the composition of this complex in mammalian neurons. PCAF (gene name *Kat2b*) is among the subunits that are down-regulated in the spinal cord tissue. However, since most of the subunits of the ATAC complex are also down-regulated (and PCAF is also present in this complex), the lower levels of PCAF may represent, in part, a coordinated down-regulation for all the subunits of the ATAC complex.

*5) Due to the possible role of Taf9b on late stages of MN differentiation, and the misregulation of several “patterning” and “axon guidance” genes based on GO terms, it could be informative to compare Taf9b with Hoxc9 binding. Hoxc9 controls patterning and the MN axon guidance in the thoracic spinal cord (PMID:20826310). The authors might consider exploring whether there is a correlation with Hoxc9, which could strengthen the concept of Taf9b binding with sequence-specific transcription factors to control CNS patterning*.

We have compared the TAF9B ChIP-seq binding sites to the HOXC9 ChIP-seq binding sites, in the same manner as we compared to OLIG2 ChIP-seq peaks, and found that only a limited number of TAF9B peaks (∼6% of TAF9B peaks) showed co-localization with HOXC9 ChIP-seq peaks (examples in Figure 5—figure supplement 1 and data not shown). Further experiments will be necessary to identify other transcriptional activators that may have strong co-localizations with TAF9B in this developmental pathway.